EMBO
Molecular Medicine

# REST suppression mediates neural conversion of adult human fibroblasts via microRNA-dependent and -independent pathways

Janelle Drouin-Ouellet[1,†], Shong Lau[1,†], Per Ludvik Brattås[1], Daniella Rylander Ottosson[1], Karolina Pircs[1], Daniela A Grassi[1], Lucy M Collins[2], Romina Vuono[2], Annika Andersson Sjöland[3], Gunilla Westergren-Thorsson[3], Caroline Graff[4,5], Lennart Minthon[6], Håkan Toresson[6], Roger A Barker[1,2], Johan Jakobsson[1] & Malin Parmar[1,*]

## Abstract

Direct conversion of human fibroblasts into mature and functional neurons, termed induced neurons (iNs), was achieved for the first time 6 years ago. This technology offers a promising shortcut for obtaining patient- and disease-specific neurons for disease modeling, drug screening, and other biomedical applications. However, fibroblasts from adult donors do not reprogram as easily as fetal donors, and no current reprogramming approach is sufficiently efficient to allow the use of this technology using patient-derived material for large-scale applications. Here, we investigate the difference in reprogramming requirements between fetal and adult human fibroblasts and identify REST as a major reprogramming barrier in adult fibroblasts. Via functional experiments where we overexpress and knockdown the REST-controlled neuron-specific microRNAs miR-9 and miR-124, we show that the effect of REST inhibition is only partially mediated via microRNA up-regulation. Transcriptional analysis confirmed that REST knockdown activates an overlapping subset of neuronal genes as microRNA overexpression and also a distinct set of neuronal genes that are not activated via microRNA overexpression. Based on this, we developed an optimized one-step method to efficiently reprogram dermal fibroblasts from elderly individuals using a single-vector system and demonstrate that it is possible to obtain iNs of high yield and purity from aged individuals with a range of familial and sporadic neurodegenerative disorders including Parkinson's, Huntington's, as well as Alzheimer's disease.

**Keywords** adult human dermal fibroblasts; induced neurons; microRNAs 9/9* and 124; RE1-silencing transcription factor

**Subject Categories** Chromatin, Epigenetics, Genomics & Functional Genomics; Neuroscience; Stem Cells

## Introduction

New advances in somatic cell reprogramming offer unique access to human neurons from defined patient groups for modeling neurological disorders *in vitro*. This has enabled a number of mechanistic studies to better understand how pathology arises and develops, and also creates new opportunities for early and differential diagnostic tests and drug screens (Kondo *et al*, 2013; Young *et al*, 2015; Mertens *et al*, 2016). The most common route to patient- and disease-specific neurons to date is through reprogramming of somatic cells into induced pluripotent stem cells (iPSCs), followed by directed neural differentiation (Nityanandam & Baldwin, 2015). Although this approach has led to important insights into neurodevelopmental disorders and mechanisms underlying neural pathologies (Ebert *et al*, 2009; Lee *et al*, 2009; Lafaille *et al*, 2012), a number of studies show that reprogramming into pluripotency resets the age of the cells such that the resulting neurons are very young (Maherali *et al*, 2007; Meissner *et al*, 2008; Lapasset *et al*, 2011; Miller *et al*, 2013; Mertens *et al*, 2015). Consequently, this approach may not be ideal for modeling all aspects of age-related neurodegenerative disorders such as Alzheimer's disease (AD), Parkinson's disease (PD), and Huntington's disease (HD).

As an alternative for generating disease- and patient-specific neurons, adult fibroblasts can be directly converted into functional

---

1  Division of Neurobiology and Lund Stem Cell Center, Department of Experimental Medical Science, Wallenberg Neuroscience Center, Lund University, Lund, Sweden
2  John van Geest Centre for Brain Repair & Department of Neurology, Department of Clinical Neurosciences, University of Cambridge, Forvie Site, Cambridge, UK
3  Department of Experimental Medical Science, Unit of Lung Biology BMC, C12 Lund University, Lund, Sweden
4  Division for Neurogeriatrics, Department of NVS, Center for Alzheimer Research, Karolinska Institutet, Huddinge, Sweden
5  Department of Geriatric Medicine, Karolinska University Hospital, Stockholm, Sweden
6  Clinical Memory Research Unit, Department of Clinical Sciences Malmö, Lund University, Lund, Sweden
   *Corresponding author. Tel: +46 46 222 06 20; E-mail: malin.parmar@med.lu.se
   †These authors contributed equally to this work

neurons using chemicals, defined sets of transcription factors or microRNAs (miRNAs) (Ambasudhan *et al*, 2011; Caiazzo *et al*, 2011; Pang *et al*, 2011; Pfisterer *et al*, 2011b; Victor *et al*, 2014; Hu *et al*, 2015). This type of direct reprogramming allows fibroblasts to be converted into induced neurons (iNs) without transitioning via a proliferative stem cell intermediate (Ambasudhan *et al*, 2011; Yoo *et al*, 2011; Fishman *et al*, 2015), making the process faster and easier. In addition, recent studies have also demonstrated that the resulting iNs, unlike iPSCs, maintain the aging signature of the donor, making iNs ideal candidates for modeling neuronal pathology in late-onset diseases (Mertens *et al*, 2015; Huh *et al*, 2016). However, several factors such as species and age of donor, passage number, and prolonged culturing of cells prior to conversion limit the reprogramming efficiency of this approach. In particular, human cells are harder to reprogram than rodent cells (Caiazzo *et al*, 2011; Xue *et al*, 2013, 2016), cells from adult donors are much harder to reprogram than fetal cells (Pfisterer *et al*, 2011b; Liu *et al*, 2013), and *in vitro* expansion and/or extensive culturing and passaging of cells prior to reprogramming prevents successful conversion (Price *et al*, 2014; Masserdotti *et al*, 2015). The reason for these differences is not fully understood, but the fact that human fibroblasts from aged individuals are more resistant/refractory to reprogramming than fetal fibroblasts creates a barrier for using these cells for large-scale biomedical applications and future clinical applications.

In this study, we performed comparative global gene expression analysis of fetal and adult fibroblasts to investigate the transcriptional response in the early stage of neural conversion to better understand the reprogramming requirements specific to adult dermal fibroblasts. From this dataset, we identified the RE1-silencing transcription factor (REST) complex as a potential barrier to reprogramming of adult human fibroblasts. We confirm this by showing that REST inhibition (RESTi), when combined with the neural conversion genes *Ascl1* and *Brn2*, can remove the reprogramming barrier in adult dermal and lung fibroblasts and yield a high number of functionally mature neurons. Via functional experiments where we overexpress or knockdown the neuron-specific miRNAs miR-9 and miR-124, we could show that the effect of RESTi during

conversion of adult fibroblasts is mediated in part via miRNA up-regulation, but also through miRNA-independent mechanisms.

Based on these data, we constructed an all-in-one neural conversion vector that contains all the components necessary for robust, high-yield neural conversion of adult dermal fibroblasts. We then demonstrated that such a vector could be used to efficiently convert fibroblasts collected at three different clinical sites from individuals with idiopathic as well as genetic forms of PD and AD as well as patients with HD. This new approach to iN conversion reported here has great potential for disease modeling across a range of neurological disorders that develop later in life—a set of conditions that until now has been nearly impossible to model using this approach.

## Results

### Development of a bicistronic vector for co-delivery of neural conversion genes

To achieve a highly effective and reproducible conversion system with less variability in transcription factor expression in each cell, we generated and tested three different dual-promoter vectors (Stadtfeld *et al*, 2010; Carey *et al*, 2011). Although the level of expression of each transgene may vary between each cell, this dual-vector approach insures a delivery of the two neural conversion genes *Ascl1* and *Brn2* in all cells. All vectors are based on the human PGK promoter, but the conversion genes were placed in a different order and distance from the woodchuck hepatitis virus posttranscriptional regulatory elements (WPRE) (Fig 1A). When expressed in human fetal fibroblasts, the three constructs resulted in different levels of expression of the conversion genes (Fig 1B and C), and we found that the pB.pA construct, yielding the highest ASCL1 to BRN2 protein expression ratio, resulted in the highest level of neural conversion (Fig 1D). However, since immunochemical staining depends on the quality of the antibody and is not quantitative, in a separate experiment, we used GFP as a reporter and placed it in two different positions in our vector (Appendix Fig S1A), and by measuring endogenous GFP expression, we confirmed that the gene placed

**Figure 1. Bicistronic approach successfully reprograms fetal fibroblasts but fails to reprogram adult fibroblasts.**

A Vector maps of constructs containing the neural conversion factors *ASCL1* coding for MASH1 and *BRN2* as well as woodchuck hepatitis posttranscriptional element (WPRE) at different positions.

B Quantitative analysis showing the difference in fluorescence intensity of ASCL1 (red bar graphs) and BRN2 (yellow bar graphs) following transduction with the different constructs.

C, D Representative images of double-immunofluorescent staining of ASCL1 (in green) and BRN2 (in red) (C) as well as MAP2 staining (D) showing the different expression levels of each transcription factor and the resulting neuronal conversion for each construct.

E Quantification of the number of iNs converted 12 days after transduction with either Pgk.Ascl1 + Pgk.Brn2 + Pgk.Myt1L or pB.pA.

F RNA-seq analysis illustrating the fold changes in gene expression in fetal fibroblasts transduced with pB.pA as compared to untransduced cells, with genes that are significantly up- or down-regulated marked as red dots.

G Gene ontology enrichment analysis reveals significant enrichment of neuronal genes (in bold) among the up-regulated genes in the pB.pA-transduced fetal fibroblasts.

H Representative fluorescence images showing the MAP2 expression in fetal and adult fibroblasts (dermal and lung) reprogrammed with pB.pA.

I FC correlation analysis and Venn diagram showing genes that are significantly changed in both adult and fetal pB.pA-transduced cells (red) and significantly changed in fetal cells only (blue) or adult cells only (green) or not changed (black).

J Gene ontology enrichment analysis showing the genes associated with neurons (in bold) that are up-regulated in the pB.pA-transduced fetal fibroblasts but not in the adult fibroblasts transduced with pB.pA.

Data information: Scale bars, 100 μm in (D), 50 μm in (H). ahDF, adult human dermal fibroblasts; ahFL, adult human lung fibroblasts; CTR, control. Data are expressed as mean ± SEM and are from biological replicates (*n* = 3). *$P < 0.05$. Exact *P*-values and statistical tests used to calculate them are provided in Appendix Table S4.

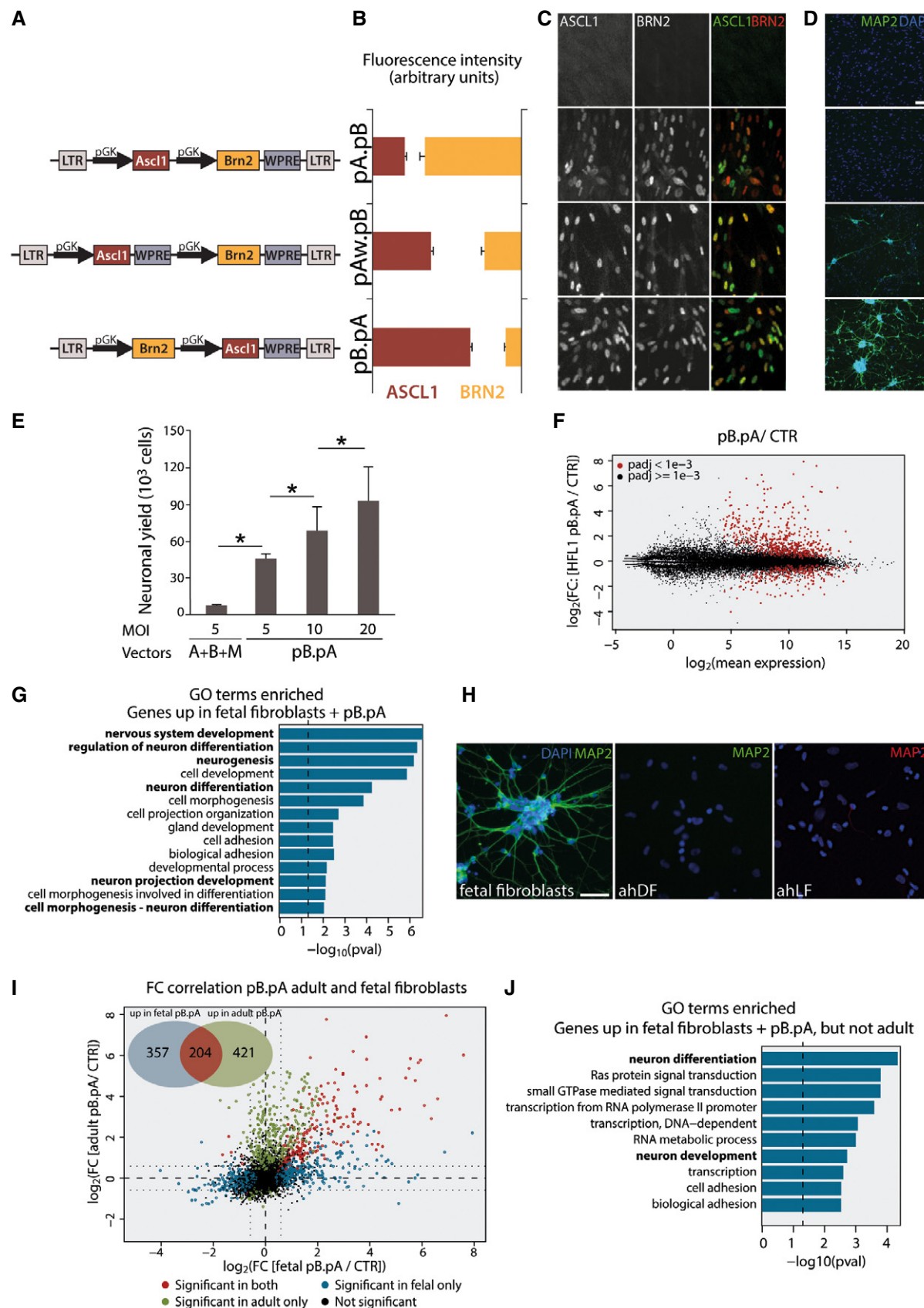

**Figure 1.**

under control of the second promoter in this construct is expressed at higher levels and in a greater number of cells (Appendix Fig S1B–D). When co-delivering the two conversion factors using the pB.pA dual-promoter vector, we found that we increased the yield of iNs by more than 30-fold compared to when the neural conversion factors were delivered using separate vectors (Fig 1E), and by increasing the viral titer, we could further increase the yield to very high levels, reaching conversion efficiencies up to 150% (i.e., 150,000 iNs generated per 100,000 fibroblasts plated, Fig 1E).

### Difference in conversion mechanism/requirement between fetal and adult fibroblasts

Global gene expression analysis confirmed that the pB.pA dual-promoter construct induced a major change in gene expression in the fetal fibroblasts. We found 561 significantly (Benjamini–Hochberg (BH)-corrected $P$-value < 0.001) up-regulated and 328 significantly down-regulated genes (Fig 1F) 5 days after delivering the conversion vector. Gene ontology analysis showed that many of the up-regulated genes were associated with a neuronal identity (Fig 1G), in line with the high conversion yield observed using this reprogramming vector. We next used the same system to convert adult human dermal fibroblasts from a healthy 67-year-old individual. However, we detected only very few, if any, iNs after 30 days (Fig 1H). To rule out the possibility that this failure to reprogram was in fact related to adult versus fetal fibroblasts and not due to difference in the origin of the fibroblast, we confirmed the failure to reprogram using adult lung fibroblasts from a 45 to 65 individual (Fig 1H).

To better understand the difference in reprogramming requirements between fetal and adult fibroblasts, we assessed the transcriptional response in the cells after delivery of the dual-conversion vector using RNA-seq. We found that while 204 genes were up-regulated ($P < 0.001$) in both adult and fetal fibroblasts after transduction with pB.pA dual-promoter vector, another 357 and 421 genes were uniquely up-regulated in the transduced fetal or adult fibroblasts, respectively (Pearson correlation: 0.307, Fig 1I). GO analysis of the genes up-regulated in the fetal, but not adult fibroblasts resulted in gene categories associated with neuronal functions (Fig 1J). This demonstrates that the neural conversion factors activate a largely different set of genes with limited overlap in the two starting populations, and suggests that there are specific barriers to reprogramming present in adult but not fetal fibroblasts. When looking at the top 11 genes related to neuronal differentiation and development uniquely up-regulated in the fetal fibroblasts, four were identified as REST targets: *JAG2, L1CAM, DYNLL2, and DCLK1,* suggesting that REST blocks the activation of neuronal genes and subsequent neuronal conversion in the adult fibroblasts.

### REST inhibition removes neural reprogramming block in human adult lung and dermal fibroblasts

To test the hypothesis that REST prevents neural conversion of adult fibroblasts transduced with ASCL1 and BRN2, we performed qRT–PCR analysis in fetal and adult fibroblasts which revealed slightly increased levels of *REST* transcripts in adult cells (Fig 2A, $P < 0.05$). We next used RNAi to knockdown REST, which reduced *REST* transcript levels in adult fibroblasts down to that observed in fetal

human fibroblasts (Fig 2A). When we expressed the dual-promoter conversion vector together with the shRNAs against *REST* in adult dermal fibroblasts from two different donors (age 61 and 67), we consistently observed exceptionally high neural conversion levels (Fig 2B). We also confirmed that RESTi removes the reprogramming barrier also of adult lung fibroblasts (Fig 2B). The high conversion efficiency was confirmed using five primary lines from dermal biopsies of individuals aged from 61 to 71 years and sourced from three different clinical sites (Fig 2C). We also observed that in contrast to previous reports demonstrating that the reprogramming efficiency decreases at higher passages (Pfisterer *et al*, 2011a; Tocchini *et al*, 2014), there was no decrease in the conversion efficiency or neuronal purity when the fibroblasts from a 67-year-old donor were reprogrammed with the dual-promoter construct and RESTi at passages ranging from 3 to 10 (Fig 2D). This implies that RESTi also removes the barriers to reprogramming associated with extensive passaging of the fibroblasts previously observed (Price *et al*, 2014; Masserdotti *et al*, 2015).

We next analyzed the mature neuronal properties of the resulting iNs. We found that they did indeed express mature neuronal markers such as MAP2, NEUN, SYNAPSIN, and TAU (Fig 2E). Patch-clamp electrophysiological recordings of the iNs after terminal differentiation and maturation in culture showed that they had acquired the functional properties of neurons (Fig 2F and Appendix Table S1). This was also the case when cells pre-labeled with a vector containing GFP expressed under the control of the human synapsin promoter were transplanted to the neonatal brain and analyzed after 7–9 weeks of maturation *in vivo*. When analyzing the transplanted iNs detected based on GFP expression, we again found current evoked multiple action potentials in the iNs ($n = 8$ from four different rats) (Fig 2G), and the cells displayed postsynaptic currents that could be blocked with the glutamate antagonist CNQX (Fig 2G), demonstrating that adult iN cells converted in the presence if RESTi functionally mature, integrate, and receive glutamatergic synaptic inputs from the host brain.

### RESTi results in up-regulation of neural-specific miRNAs

MiRNAs have been implicated as important mediators of cell reprogramming (Adlakha & Seth, 2017), including in neural conversion (Yoo *et al*, 2011; Xue *et al*, 2013, 2016; Victor *et al*, 2014). Inhibition of REST is known to increase expression of neuron-specific miRNAs (Ballas *et al*, 2005; Conaco *et al*, 2006), and we speculated that the potential up-regulation of miRNAs could be what mediated the effect of RESTi during neural conversion of adult human fibroblasts. We therefore assessed the neuron-specific miRNA expression levels in the absence and presence of RESTi and found that miR-9 was up-regulated when adult fibroblasts are converted in the presence of RESTi (Fig 3A). We also checked the expression of several region-specific miRNAs (Jönsson *et al*, 2015) but found no clear differences, indicating that RESTi affects pan-neuronal expression without affecting subtype identity (Fig 3B). To further investigate this, we tested whether expression of neuron-specific miRNAs could mimic the effect of RESTi. We therefore expressed miR-9/9* and miR-124 together with the conversion factors (Fig 3C) but without RESTi. We found that adult fibroblasts transduced with this construct expressed high levels of miR-9 and miR-124 (Appendix Fig S2A and B) and converted adult fibroblasts into neurons with

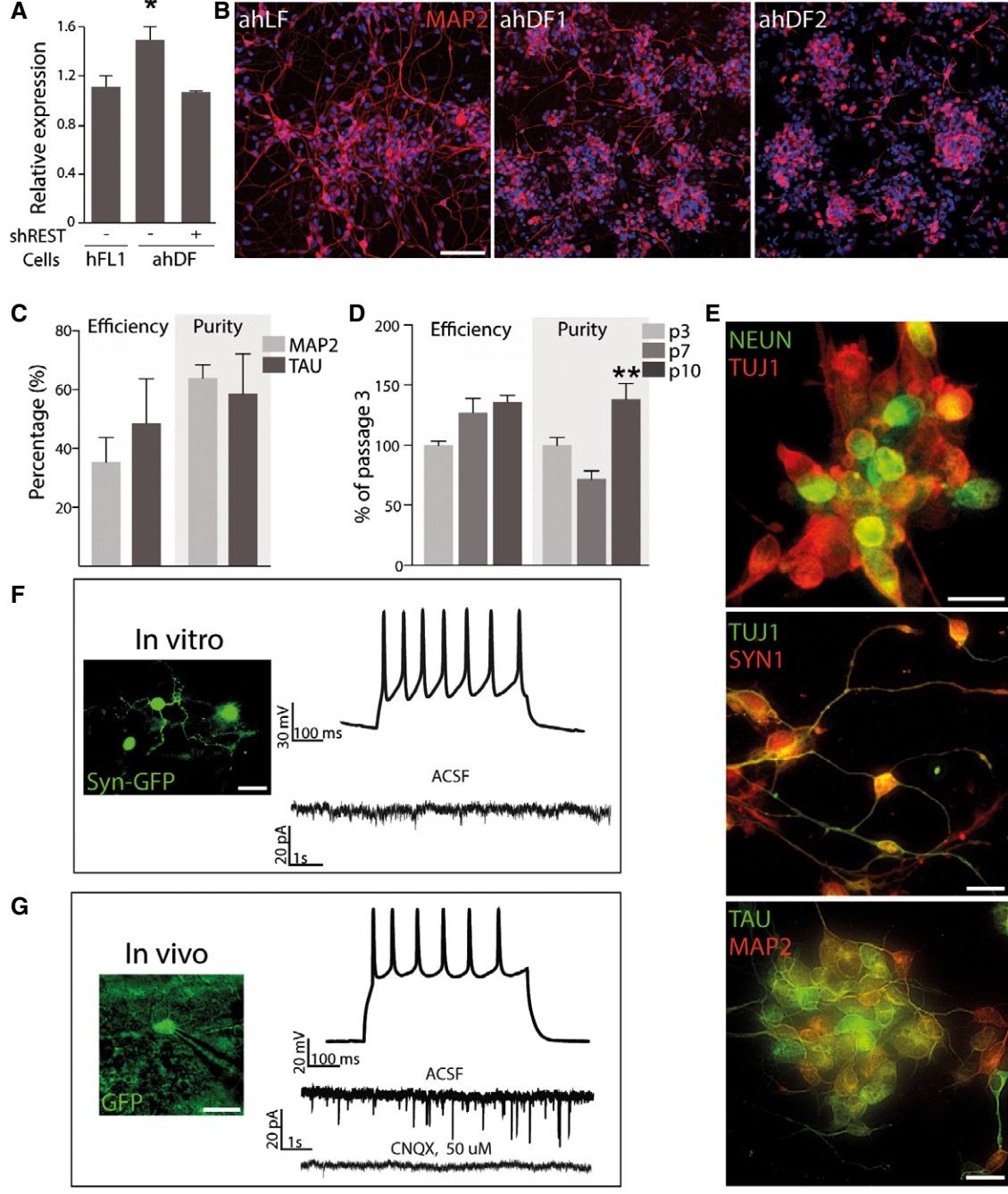

**Figure 2.  *REST* knockdown promotes the pB.pA-driven reprogramming of adult human fibroblasts.**

A    qPCR analysis of *REST* gene expression.
B    Representative immunofluorescence images showing a high density of MAP2⁺ cells in pB.pA + RESTi reprogrammed adult fibroblasts from different sources.
C    Quantification of neuronal efficiency and purity of pB.pA + RESTi reprogrammed adult human dermal fibroblasts from five healthy donors (61–71 years).
D    Quantification of neuronal efficiency and purity of an adult human dermal fibroblast line reprogrammed with pB.pA + RESTi at different passages.
E    Double-immunofluorescence stainings showing the expression of neuronal markers in iNs reprogrammed from adult fibroblasts 25 days post-transduction with pB.pA + RESTi.
F    *In vitro* patch-clamp recordings of adult iNs depicting repetitive current-induced action potentials indicative of mature neuronal physiology at 12–15 weeks post-transduction.
G    Presence of repetitive current-induced action potentials and spontaneous postsynaptic currents *in vivo* 8 weeks following transplantation.

Data information: Scale bars, 100 μm in (B), 25 μm in (E–G). ahDF, adult human dermal fibroblasts; ahFL, adult human lung fibroblasts; shREST, short hairpin RNA against *REST*. Data are expressed as mean ± SEM and are from biological replicates (n = 3–4). *P < 0.05, **P < 0.01. Exact P-values and statistical tests used to calculate them are provided in Appendix Table S4.

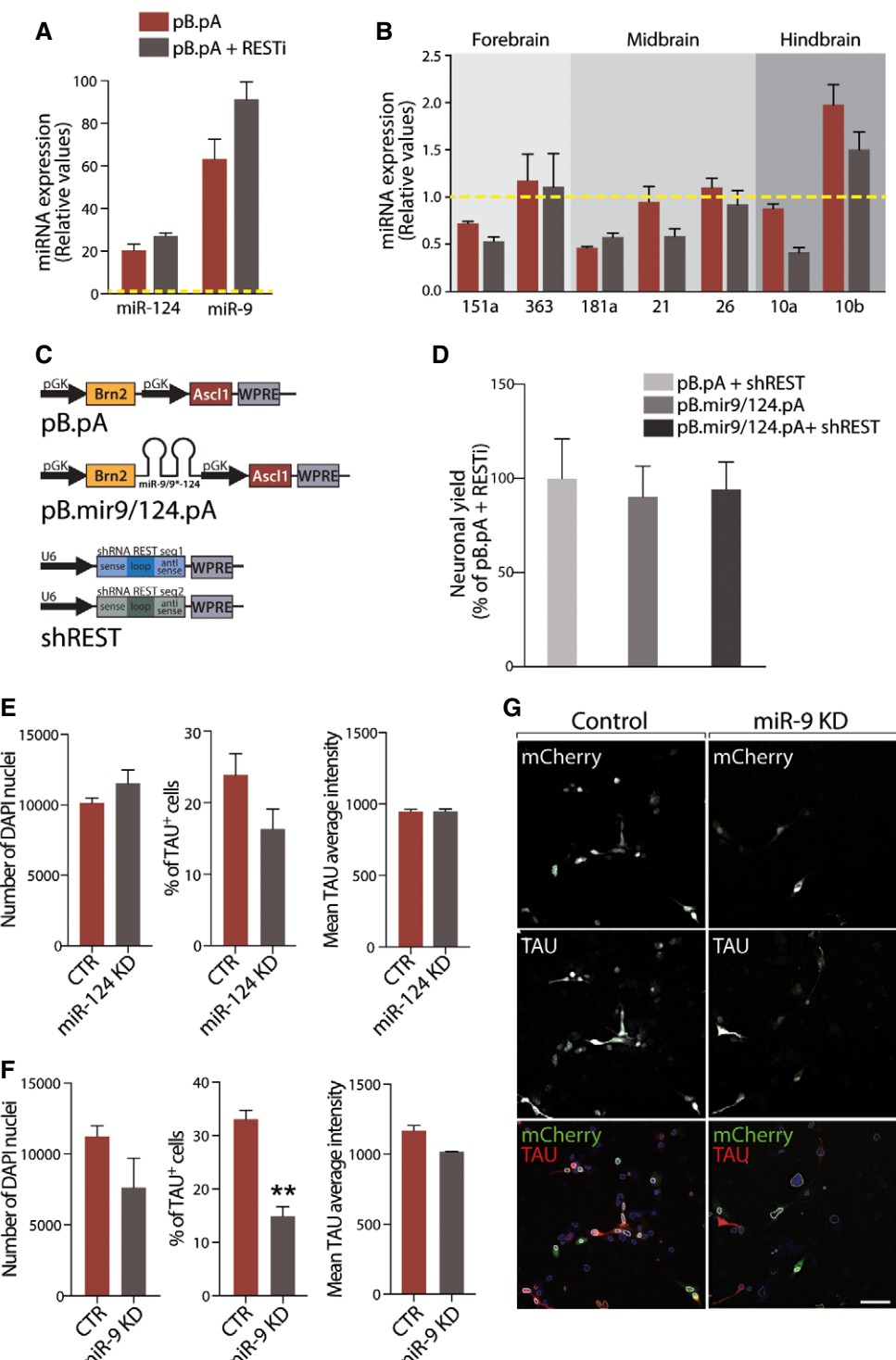

Figure 3.

similar efficiency to the cells treated with RESTi (Fig 3D), supporting the hypothesis that RESTi effect could be mediated via up-regulation of miR-9/9* and miR-124, and that miRNAs, like RESTi removes the reprogramming barrier in adult fibroblasts allowing also fibroblasts from aged donors to efficiently and reproducibly be converted into neurons.

To experimentally address whether the RESTi effect is mediated via miRNA up-regulation, we next performed conversions using pB.pA + RESTi while simultaneously knocking down miR-124 or miR-9 in the cells and checked for effects on neural conversion (Fig 3E–G). We found that while inhibition of miR-124 during the conversion did not significantly affect the iN conversion (Fig 3E),

**Figure 3. Neuronal microRNA expression partly drives neuronal reprogramming of adult fibroblasts.**

A    qPCR measurements of miR-124 and miR-9 in adult fibroblasts reprogrammed with pB.pA only or pB.pA + RESTi and normalized on the non-transduced fibroblast values (yellow dashed line).

B    Region-specific microRNAs qPCR measurements in adult fibroblasts reprogrammed with pB.pA only or pB.pA + RESTi and normalized on the non-transduced fibroblast values (yellow dashed line).

C    Vector maps of constructs containing the transcription factors Ascl1 and Brn2 with and without miR-9 and miR-124, as well as the shRNA sequences against *REST*.

D    Quantification of the neuronal yield as assessed by MAP2 expression in adult fibroblasts transduced with different reprogramming vectors.

E, F    Quantification of the total number of cells as well as the percentage of TAU⁺ cells and the average fluorescence intensity in adult iNs with and without miR-124 (E) or miR-9 (F) knockdown.

G    Representative images of the high content screening target activation analysis showing the cells expressing mCherry (successfully transduced with the miRNA inhibition or control constructs) that have been included in the analysis of the TAU staining (white contours). Rejected nuclei are circled in yellow and valid nuclei that do not express mCherry exhibit blue contours.

Data information: Scale bar, 50 μm in (G). CTR, control; KD, knockdown. Data are expressed as mean ± SEM and are from biological replicates ($n$ = 3–4). **$P$ < 0.01. Exact $P$-values and statistical tests used to calculate them are provided in Appendix Table S4.

the inhibition of miR-9 during the reprogramming resulted in a decrease in the number of iNs generated compared to control (Fig 3F and G).

Taken together, our data show that the effect of RESTi can be mimicked via miRNA overexpression but that blocking miRNA inhibition during the conversion process only partially affects the neural conversion. This supports that the RESTi acts, at least partially, via miRNA activation and via the previously suggested interplay between RESTi and miRNAs (Ballas *et al*, 2005; Conaco *et al*, 2006; Xue *et al*, 2013, 2016).

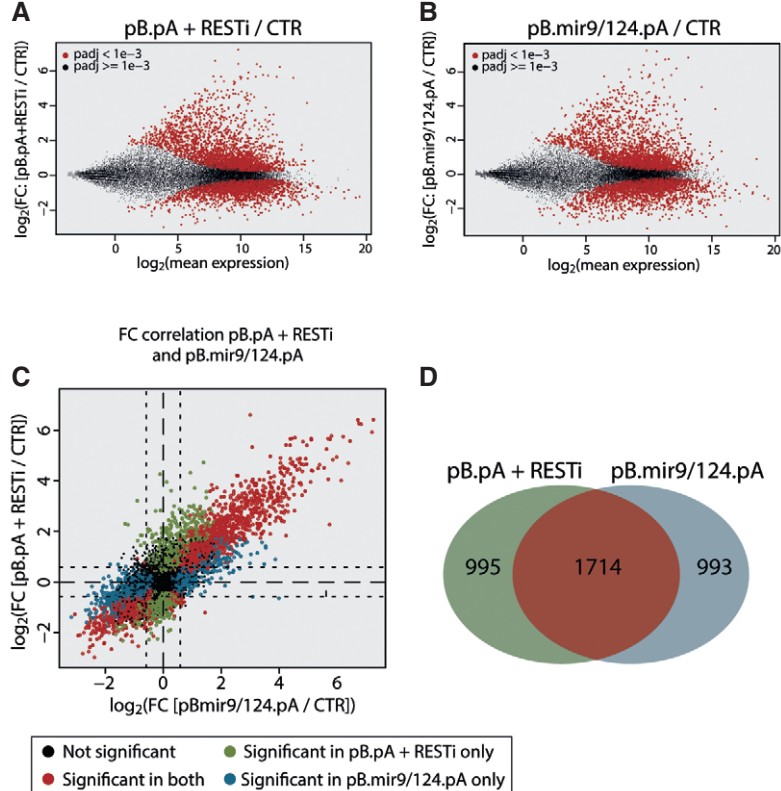

**Figure 4. Differences in gene expression between pB.pA + RESTi and pB.mir9/124.pA.**

A    Graph illustrating the fold changes in gene expression in adult fibroblasts transduced with pB.pA + RESTi as compared to untransduced cells (genes that are significantly up- or down-regulated marked as red dots).

B    Graph illustrating the fold changes in gene expression in adult fibroblasts transduced with pB.mir9/124.pA as compared to untransduced cells (genes that are significantly up- or down-regulated marked as red dots).

C    FC correlation analysis showing the genes that are significantly changed in both pB.pA + RESTi- and pB.mir9/124.pA-transduced cells (red) that are significantly changed in pB.mir9/124.pA-transduced cells only (blue) or pB.pA + RESTi cells only (green) or unchanged (black).

D    Venn diagram showing the genes that are significantly changed in both pB.pA + RESTi- and pB.mir9/124.pA-transduced cells (red) that are significantly changed in pB.mir9/124.pA-transduced cells only (blue) or pB.pA + RESTi cells only (green).

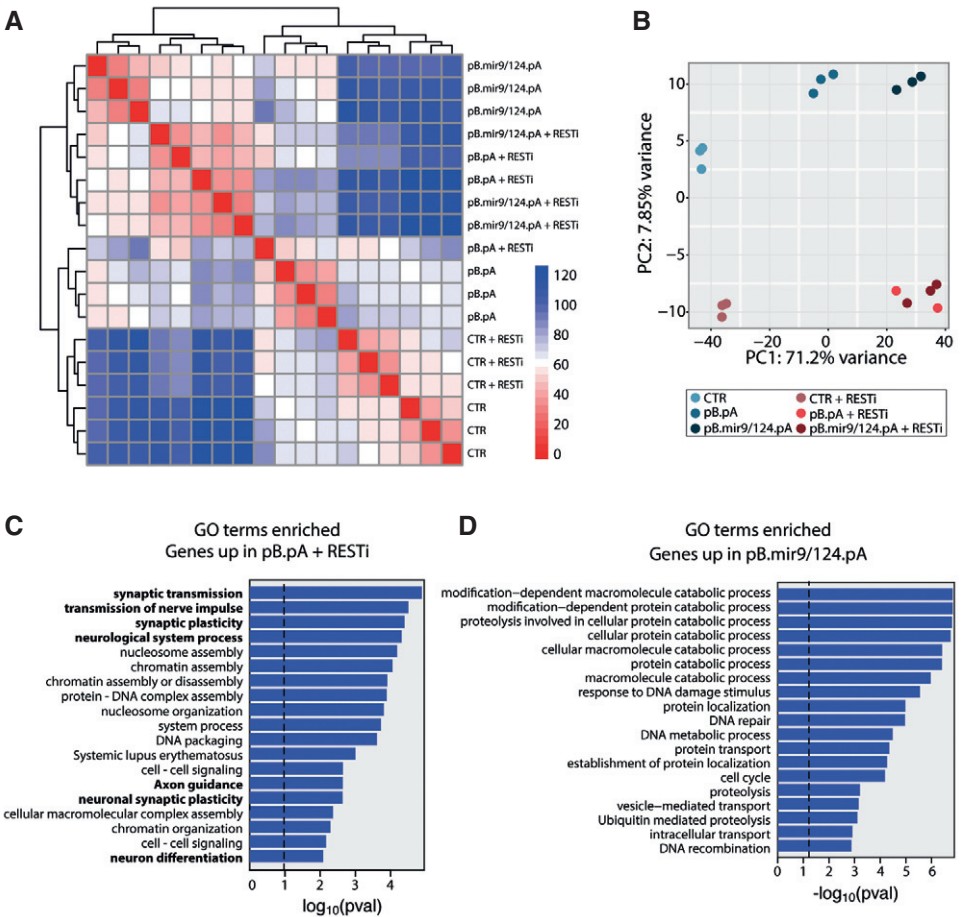

**Figure 5. Enhanced neuronal gene up-regulation in pB.pA + RESTi reprogrammed adult iNs.**

A  Clustering of RNA-seq samples, using euclidean distance on normalized and log-transformed read counts.
B  Principal component analysis showing a separation of the groups that reprogram well from those that do not on PC1 and a separation of the groups with *REST* knockdown on PC2.
C  Gene ontology enrichment analysis showing significant enrichment of neuronal genes (in bold) among the genes up-regulated in the pB.pA + RESTi-transduced fibroblasts but not in the adult fibroblasts transduced with pB.mir9/124.pA.
D  Gene ontology enrichment analysis showing that no genes associated with neurons are uniquely up-regulated in the pB.mir9/124.pA-transduced fibroblasts.

## MicroRNA-independent effects of REST inhibition

To better understand the mechanisms that mediate the conversion of adult fibroblasts driven by RESTi or miR-9/miR-124, we performed a comparative global gene expression analysis using RNA sequencing 5 days following the initiation of conversion. In this analysis, we included unconverted adult human fibroblasts and adult fibroblasts in which REST is knocked down as controls. The conversion groups included were as follows: pB.pA (that gives rise to only very low-level iN conversion if any); pB.pA + RESTi; pB.miR9/124.pA; and pB.miR9/124.pA + RESTi. We compared the genes up-regulated (BH-corrected *P*-value < 0.001) in the pB.pA + RESTi group and the pB.miR9/124.pA groups. This analysis showed that both RESTi and miR-9/miR-124 delivery caused a major transcriptomic change in the cells and that the effect was not cumulative (Fig 4A and B). Further analysis showed that most of the genes with the largest FC are significant in both the miR-9/miR-124- and RESTi-transduced cells (Pearson correlation = 0.81, Fig 4C

and D). Most genes (more than 1700) were up-regulated in both groups suggesting that these factors largely work on the same neurogenic pathway(s) and activate similar gene cascades.

We next investigated in more detail the differences in gene expression profiles between the RESTi- and miRNA-converted cells. Unsupervised clustering revealed that the two controls (fibroblasts and fibroblasts + RESTi) as well as the pB.pA (very low conversion group) clustered together, while all three groups with successful neural conversion clustered together (Fig 5A). Principal component analysis revealed that the three conversion groups were very similar on the PC1 axis and distinctly different from the control groups. Furthermore, the PC2 axis showed a separation of the groups with RESTi from those without (Fig 5B). The GO term and Kyoto Encyclopedia of Genes and Genomes (KEGG) pathway analyses of the differentially expressed genes revealed that those differentially expressed in the RESTi conversion group were enriched for the regulation of synaptic transmission, synaptic plasticity, as well as cell morphogenesis and the differentiation and regulation of

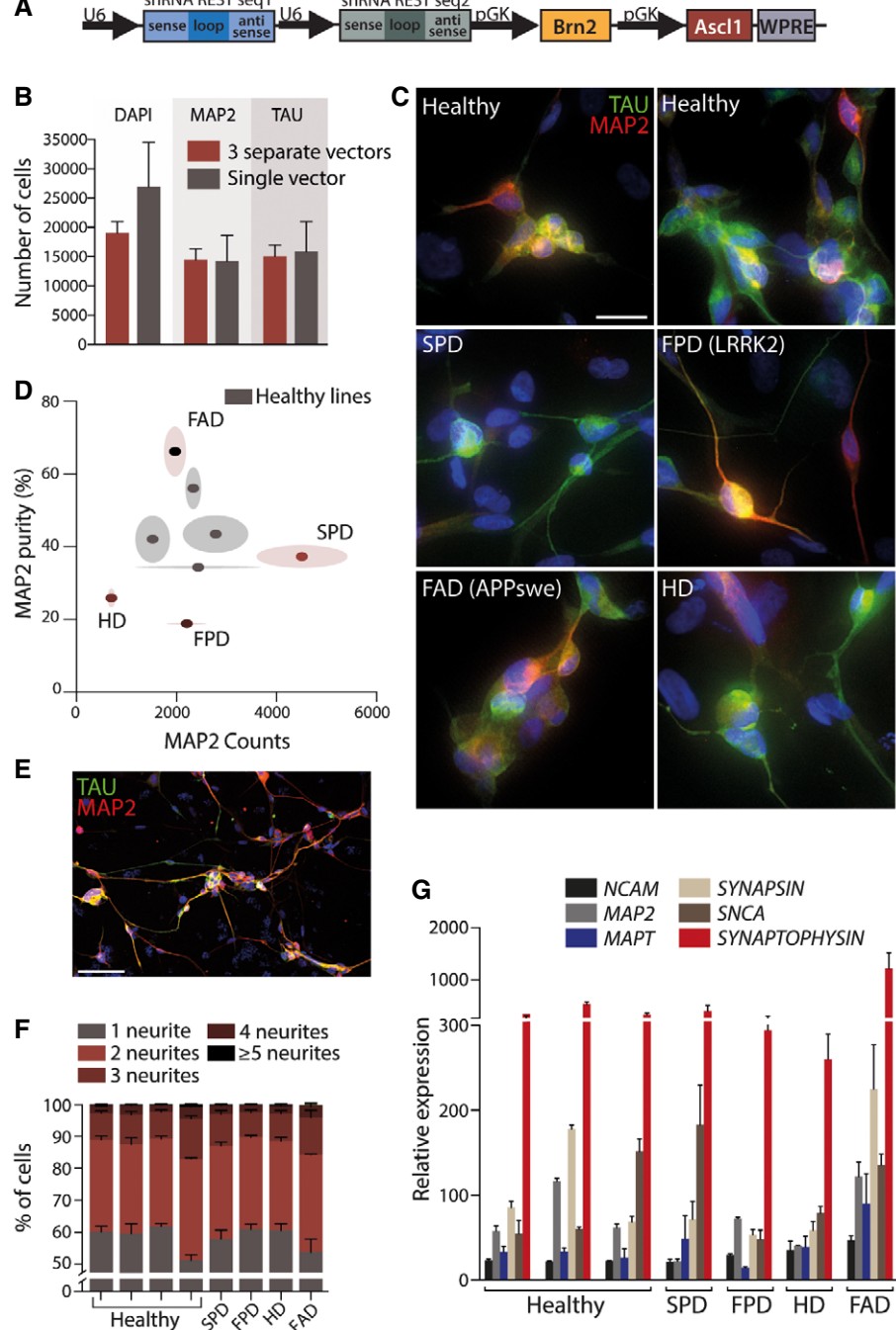

**Figure 6.  All-in-one vector to reprogram skin fibroblasts from patients with a range of different neurodegenerative disorders.**

A   Map of the single reprogramming vector containing *REST* shRNA sequences as well as *Brn2* and *Ascl1*.
B   Quantitative comparison of the total number of cells, as well as the number of MAP2+ and TAU+ cells per well using separate vectors or one single vector for pB.pA + RESTi reprogramming in four different adult dermal fibroblast lines.
C   Fluorescence microscopy images of iNs reprogrammed using the single vector from healthy individuals as well as from patients with various neurodegenerative disorders.
D   Quantification of the neuronal counts and purity.
E   Fluorescence microscopy images of iNs derived from an HD patient after optimization of culture conditions.
F   Percentage of cells displaying various number of neurites for each line.
G   qPCR analysis of six neuronal genes in healthy individuals as well as from patients with various neurodegenerative disorders.

Data information: Scale bars, 25 μm in (C), 100 μm in (E). FAD, familial Alzheimer's disease; FPD, familial Parkinson's disease; HD, Huntington's disease; SPD, sporadic Parkinson's disease. Data are expressed as mean ± SEM and are from biological replicates (*n* = 4).

neurogenesis and synapse formation (Fig 5C). In contrast, the genes uniquely up-regulated in the pB.miR9/124.pA were not associated with neuronal properties (Fig 5D).

Taken together, our results show that the RESTi, when combined with the neural conversion genes *Ascl1* and *Brn2*, overcomes human-specific barriers of both reprogramming and neuronal maturation. The miRNA knockdown experiments, as well as the global transcriptome analysis, suggest that this effect is only partially mediated via miR-9/miR-124 expression.

Based on this, we designed and cloned a single "all-in-one" construct that expressed both RESTi hairpins and conversion genes on the same construct (Fig 6A). This vector resulted in similar conversion efficiencies compared to the vector system in which the conversion genes are delivered using the dual-promoter vector pB.pA and the two REST shRNAs on three separate vectors (Fig 6B). Modeling neurodegenerative disorders would greatly benefit from this technology, as iNs from elderly donors have been shown to maintain their aging signature, which is critical given that age is the biggest risk factor for developing these disorders. To establish its utility for generating cells for disease modeling, we used the new single-vector system to convert dermal fibroblasts from healthy adults as well as individuals with sporadic PD, familial PD (LRRK2 c.6055G>A mutation), HD (41 CAG repeats), and familial AD (APP KM670/671NL mutation) (Appendix Table S2). All lines were successfully converted to iNs expressing MAP2 (Fig 6C), albeit with some variation between the lines in terms of yield and purity (Fig 6D). In particular, the HD and familial PD showed less conversion than other lines, but line-specific optimization of cell culture condition where cell passaging was omitted was able to increase the conversion as exemplified by the HD line (MAP2$^+$ neuronal count 13,857 $\pm$ 250; purity 62.4 $\pm$ 3.4%) (Fig 6E). We also used TAU as a neuronal marker in addition to MAP2 to assess the conversion into more mature neurons. Conversion of all lines resulted in neurons with a similar morphological complexity as assessed by the proportion of cells developing variable numbers of neurites for each line (Fig 6F). Additionally, qPCR analysis revealed a major increase in all the neuronal genes that we assessed (*NCAM, MAP2, MAPT, SYNAPSIN, SNCA,* and *SYNAPTOPHYSIN)* in every line converted, independently of the disease status of the donor (Fig 6G).

## Discussion

The direct conversion of one cell type to another, without going through a stem cell intermediate, has been successfully achieved for several cell types including the generation of neurons. This type of conversion makes it possible to study otherwise hard to access patient- and disease-specific neurons and holds great promise for creating age-relevant models of neurological disorders. iNs, that are obtained via direct conversion, present a faster route by which to generate neurons compared to conventional reprogramming approaches using induced pluripotent stem cells (iPSCs) followed by directed differentiation. However, as iN technology converts one mature cell type directly into a postmitotic neuron, the requirement for high-yield conversion is essential in order to obtain a sufficient number of neurons for downstream applications.

To date, a few studies have reported successful neural reprogramming of adult primary dermal fibroblasts using a wide array of conversion genes, chemical cocktails, and miRNAs, but all have resulted in relatively low numbers of induced neurons (Ambasudhan *et al*, 2011; Caiazzo *et al*, 2011; Pfisterer *et al*, 2011b; Iovino *et al*, 2014; Hu *et al*, 2015; Xu *et al*, 2015). While purification steps or antibiotic selection can increase the purity of the iNs (Vierbuchen *et al*, 2010; Victor *et al*, 2014; Mertens *et al*, 2015), this is associated with large cell loss making the yield low which in turn requires a high number of input cells which is a major drawback since adult dermal fibroblasts do not expand indefinitely. In this study, we set out to gain a better mechanistic understanding of the road blocks to reprogramming present specifically in adult human fibroblasts, by studying the early transcriptional response in fetal versus adult fibroblasts. We found that the most commonly used neural conversion genes (*ASCL1* and *BRN2*) elicit largely distinct transcriptional response in these two populations. Bioinformatics analysis confirmed that many of the genes that were up-regulated only in the fetal fibroblasts were REST targets and thus suggested REST as a potential adult-specific reprogramming barrier. This is in line with recent studies that had shown REST as a specific barrier for reprogramming of cultured astrocytes (Masserdotti *et al*, 2015) and in mouse and human fibroblasts (Xue *et al*, 2013, 2016).

We thus focused our subsequent studies on the knockdown of REST, which has been shown to release roadblocks of conversion. RESTi has also been shown to induce the expression of miR-124 as well as miR-9 in a number of cell types (Conaco *et al*, 2006; Xue *et al*, 2013) which is interesting given that these miRNAs can mediate neural conversion alone or when expressed together with neuronal transcription factors (Yoo *et al*, 2011; Xue *et al*, 2013, 2016; Victor *et al*, 2014). Our study shows that a reprogramming strategy for adult fibroblasts based on REST inhibition indeed results in increased expression of the pan-neuronal miR-9 and miR-124, while not affecting regionally expressed neural miRNAs (Jönsson *et al*, 2015). We also show that while the effect of RESTi can be partially mimicked via overexpression of neuron-specific miRNAs, inhibiting activation of miRNAs during the neural conversion process only partially inhibits the formation of iNs. This suggests that RESTi mediates its effect on neural conversion both via up-regulation of neuronal miRNAs but also via a miRNA-independent mechanism. This hypothesis was supported by our comparative RNA-seq analysis that revealed that while many of the same neuronal genes are up-regulated in fibroblasts converted with RESTi, miRNA overexpression, or both RESTi and miRNA expressions combined, additional gene transcription changes that are associated with a neuronal identity are uniquely up-regulated when fibroblasts are reprogrammed in the presence of RESTi.

Combined, our results show that a conversion strategy based on co-delivery of the conversion factors Ascl1 and Brn2 in combination with RESTi is sufficient to overcome the reprogramming barriers previously associated with adult donors, in the absence of additional miRNA expression. It results in high efficiency and high purity conversion of aged dermal fibroblasts without the need for a purification step. In addition, we also show that the passage number of the starting fibroblast culture does not impact on the reprogramming efficiency, at least up until 10 passages, ensuring that one skin biopsy will provide enough iN material to complete large-scale disease modeling, drug screening, and transplantation studies. For example, with the efficiency of our system, it would be possible to obtain ~10 billion neurons from one skin biopsy, which by far

makes our method the most efficient approach reported to date using skin biopsies from elderly donors. This makes our approach suitable to explore any potential disease-associated phenotypes in these cells, as well as offering a readily available source of relevant cells for drug screenings and diagnostics.

# Materials and Methods

### Biopsy sampling

Adult dermal fibroblasts were obtained from the Parkinson's Disease Research and Huntington's disease clinics at the John van Geest Centre for Brain Repair (Cambridge, UK) and used under local ethical approval (REC 09/H0311/88); from the Clinical Memory Research Unit (Malmö, Sweden) and used under the Regional Ethical Review Board in Lund, Sweden (Dnr 2013-402); from the Karolinska Institutet (Stockholm, Sweden) (Dnr 2005/498-31/3, 485/02; 2010/1644-32); and lung fibroblasts from a healthy individual with no clinical history of lung disease from Lunds Universitet under approval of the local Ethics committee (Dnr 413/2008 and 412/03) (see Appendix Table S2). Written informed consent was obtained from each participant, and the experiments conformed to the principles set out in the WMA Declaration of Helsinki and the Department of Health and Human Services Belmont Report. The skin biopsies were taken with a 4-mm punch biopsy from the upper or lower arm under local anesthetic (1% lidocaine), and the site was then closed with steri-strips or a stitch. Primary fibroblast from biopsies was cultured according to the two following methods: (i) Fibroblasts were isolated using standard fibroblast medium (Dulbecco's modified Eagle's medium (DMEM) + Glutamax (Gibco) with 100 mg/ml penicillin/streptomycin (Sigma), and 10% FBS (Biosera)). The skin biopsy was sectioned into 4–6 pieces and placed in a 6-cm dish coated with 0.1% gelatin containing 1.5 ml of medium, which was topped up with 0.5 ml every 2–3 days for a week. One week after the initial plating down of the cells, all of the medium was removed and 2 ml of fresh medium was added. Medium was changed every 3–4 days until full confluency of the fibroblasts was observed. The skin biopsy specimen was then transferred into a new dish, and the process was repeated until no more cells grew out of the biopsy. (ii) Subjects from the Swedish Biofinder Study had a 3-mm skin punch biopsy taken through the whole dermis to the subcutaneous fat layer using standard clinical procedures. The biopsies were immediately placed on ice in phosphate-buffered saline containing calcium and magnesium with glucose (1.8 g/l) and antibiotic–antimycotic (Gibco). Within 1.5–4 h, the biopsies were cut into 10–15 pieces avoiding the subcutaneous fat and the epidermis. The dermal pieces were placed in one well of a six-well culture plate (Nunclon) and left inside a laminar flow cabinet until dry, usually for < 15 min. 2 ml fibroblast culture medium (DMEM, 20% FBS, penicillin–streptomycin, sodium pyruvate, and antibiotic–antimycotic, all from Gibco) was then added. Incubation was in a standard cell culture incubator in 5% $CO_2$ and humidified air at 37°C. Half the medium was changed twice weekly. When ~30% of the culture well surface was covered by fibroblasts, cells were harvested by trypsinisation for ~5 min at 37°C (0.05% trypsin/EDTA, Sciencell). Cells were washed, centrifuged for 3 min at 100 × *g* at room temperature,

transferred to a T25 culture flask (Nunc), and cultured in either DMEM (as above but with 10% FBS) or in a defined serum-free medium (Fibrolife, Lifeline Celltech). The explants were fed with new DMEM with 20% FBS and placed back in the incubator to allow more fibroblasts to migrate out. Fibroblasts expanded in T25 flasks were either transferred to one T75 flask (Nunc) or frozen for long-term storage. For the lung biopsy, alveolar parenchymal specimens were collected 2–3 cm from the pleura in the lower lobes. Vessels and small airways were removed from the peripheral lung tissues, and the remaining tissues were chopped into small pieces and allowed to adhere to the plastic of cell culture flasks for 4 h. They were then kept in cell culture medium in 37°C cell incubators until the outgrowth of fibroblasts was confluent.

### Cell culture and cell lines

HFL1 (ATCC-CCL-153) cells were obtained from the American Type Culture Collection (ATCC) and expanded in standard fibroblast medium. All the fibroblasts used in this study were expanded at 37°C in 5% $CO_2$ in fibroblast medium. The cells were then dissociated with 0.05% trypsin, spun, and frozen in either 50/50 DMEM/FBS with 10% DMSO (Sigma) or DMEM + 10% FBS with 10% DMSO. Each cell line used in this study has been tested regularly for mycoplasma.

### Viral vectors and virus transduction

DNA plasmids expressing mouse open-reading frames (ORFs) for *Ascl1, Brn2,* or *Myt1L* or a combination of *Ascl1* and *Brn2* with or without short hairpin RNA (shRNA) targeting REST or miRNA loops for miR-9/9* and miR-124 were generated in a third-generation lentiviral vector containing a non-regulated ubiquitous phosphoglycerate kinase (PGK) promoter (Figs 1A, 3B and 5A). For electrophysiological recordings, a lentiviral vector expressing GFP under the control of the neuron-specific Synapsin promoter was generated and cells were transduced at a multiplicity of infection (MOI) of 5 on day 0. All the constructs have been verified by sequencing. Lentiviral vectors were produced as previously described (Zufferey *et al*, 1997) and titrated by quantitative PCR (qPCR) analysis (Georgievska *et al*, 2004). Unless otherwise stated, transduction was performed at a MOI of 10 for separate vectors and MOI 20 for the single vector (all viruses used in this study tittered between $3 \times 10^8$ and $6 \times 10^9$).

### Neural reprogramming

For direct neural reprogramming, fibroblasts were plated at a density of 27,800 cells per $cm^2$ in 24-well plates (Nunc) coated with 0.1% gelatin (Sigma). Three days after viral transduction, fibroblast medium was replaced by neural differentiation medium (NDiff227; Takara-Clontech) supplemented with growth factors at the following concentrations: LM-22A4 (2 μM, R&D Systems), GDNF (2 ng/ml, R&D Systems), NT3 (10 ng/μl, R&D Systems) and db-cAMP (0.5 mM, Sigma) and the small molecules CHIR99021 (2 μM, Axon), SB-431542 (10 μM, Axon), noggin (0.5 μg/ml, R&D Systems), LDN-193189 (0.5 μM, Axon), as well as valproic acid sodium salt (VPA; 1 mM, Merck Millipore). Half of the neuronal conversion medium was replaced every 2–3 days. Cells were

replated onto a combination of polyornithine (15 µg/ml), fibronectin (0.5 ng/µl), and laminin (5 µg/ml) coated 24-well plates at day 12 post-transduction. Eighteen days post-transduction, the small molecules were stopped and the neuronal medium was supplemented with only the growth factors (LM-22A4, GDNF, NT3, and db-cAMP) until the end of the experiment.

## microRNA knockdown experiment

Eight tandem repeats of an imperfectly complementary sequence, forming a central bulge when binding to miR-9 and miR-124 (knockdown sponge sequence), were synthesized and cloned into a third-generation lentiviral vector under a PGK promoter (see Fig 3B). The sponge sequences were as follows: miR-9 TATCATACA GCTACGACCAAAGACG and miR-124 TGGCATTCATACGTGCCTT AA. A detailed description of how to design and use lentiviral miRNA reporters and sponge vectors has been described previously (Brown *et al*, 2007; Gentner *et al*, 2009). Adult dermal fibroblasts were transduced with lentiviral vectors containing pgk.Brn2.pgk. Ascl1 (pB.pA), *REST* shRNA (all MOI = 10), and either mCherry.mir-9.sp and GFP.mir-124.sp or control vectors containing the reporter gene only (mCherry or GFP) (All MOI = 5). Cells were transduced again weekly with the mCherry.mir-9.sp, GFP.mir-124. sp, mCherry, or GFP, and triplicates of each conditions were analyzed at 25 days post-transduction with the reprogramming factors. Average fluorescence intensity analysis was performed on GFP$^+$ or mCherry$^+$ cells.

## Immunocytochemistry, imaging, and high content screening quantifications

Cells were fixed in 4% paraformaldehyde and permeabilized with 0.1% Triton X-100 in 0.1 M PBS for 10 min. Thereafter, cells were blocked for 30 min in a solution containing 5% normal serum in 0.1 M PBS. The following primary antibodies were diluted in the blocking solution and applied overnight at 4°C: mouse anti-ASCL1 (1:100, BD Biosciences, 556604), goat anti-BRN2 (1:500, Santa Cruz Biotechnology, sc-6029), rabbit anti-MAP2 (1:500, Millipore, Ab5622), mouse anti-MAP2 (1:500, Sigma, M1406), mouse anti-NEUN (1:100, Millipore, MAb377), rabbit anti-SYNAPSIN I (1:200, Millipore, 514777), mouse anti-TAU clone HT7 (1:500, Thermo Scientific, MN1000), and rabbit anti-TUJ1 (1:500, BioLegend, 802001). Fluorophore-conjugated secondary antibodies (Jackson ImmunoResearch Laboratories) were diluted in blocking solution and applied for 2 h. Cells were counterstained with DAPI for 15 min followed by three washes in PBS. The total number of DAPI$^+$, MAP2$^+$, and TAU$^+$ cells per well as well as the average fluorescence intensity for ASCL1, BRN2, and TAU were quantified using the Cellomics Array Scan (Array Scan VTI, Thermo Fischer), which is an automated process insuring unbiased measurements between groups. Applying the program "Target Activation", 289 fields (10× magnification) were acquired in a spiral fashion starting from the center. The same array was used for the analysis of the number of neurites per TAU$^+$ cells using the program "Neuronal Profiling". Neuronal purity was calculated as the number of MAP2$^+$ or TAU$^+$ over the total number of cells in the well at the end of the experiment, whereas conversion efficiency was calculated as the number of TAU$^+$ over the total number of fibroblasts plated for reprogramming.

## Fluorescence activated cell sorting

For qRT–PCR analysis of neuronal gene expression, reprogrammed cells were detached from cultureware with Accutase (PAA Laboratories), gently triturated, and washed with washing buffer containing Hank's balanced salt solution (GIBCO) with 1% bovine serum albumin and DNAse. Fibroblasts were either directly used for sorting according to GFP expression or incubated in washing buffer containing a mouse anti-human NCAM antibody labeled with APC (1:50 for fetal fibroblasts or 1:10 for adult fibroblasts, BD Biosciences) for 15 min at 4°C. The cells were sorted using a FACSAria III cell sorter according to human NCAM (neural cell adhesion molecule 1) expression gated against unstained converted iNs.

## qRT–PCR analysis for miR-9-, miR-124-, and RE1-silencing transcription factor

Total RNA, including miRNA, was extracted from human fibroblasts as well as NCAM$^+$ sorted converted fibroblasts from the same lines using the micro miRNeasy kit (Qiagen) followed by Universal cDNA synthesis kit (Fermentas, for RNA analysis; Exiqon for miRNA expression). Three reference genes were used for each qPCR analysis (ACTB, GAPDH, and HPRT1). Primer sequences can be found in Appendix Table S3. LNA–PCR primer sets, specific for hsa-miR-9-5p, hsa-miR-124-3p, and hsa-miR-103 (the latter used as normalization miRNA), were purchased from Exiqon and used for the miRNA qPCR analysis. All primers were used together with LightCycler 480 SYBR Green I Master (Roche). Standard procedures of qRT–PCR were used, and data were quantified using the $\Delta\Delta C_t$ method. Statistical analyses were performed on triplicates from each group.

## RNA-seq analysis

Fibroblasts were transduced with the different lentiviral vectors (pB.pA or pB.mir9/124.pA $\pm$ RESTi), and both untransduced fibroblasts and fibroblasts transduced only with REST shRNA were used as controls (CTR). Cells were collected 5 days after transduction. RNA was extracted using RNeasy mini kit (Qiagen) with DNase treatment and sent for RNA-seq to UCLA Clinical Microarray Core. cDNA libraries were prepared using the KAPA Stranded mRNA-Seq Kit from KAPAbiosystems. The 50-bp single-end reads from the Illumina HiSeq 2000 were mapped to the human genome assembly (GRCh38) using STAR (2.4.0j) (Dobin *et al*, 2013) with default parameters. mRNA expression was quantified using the subread package FeatureCounts (Liao *et al*, 2014) quantifying to NCBI annotation (GRCh38). Read counts were normalized to the total number of reads mapping to the genome. Clustering and differential expression analysis were done with DESeq2 (Love *et al*, 2014). Downstream analyses were performed using in-house R and unix scripts. Gene ontology analysis was done with the Functional Annotation Tool of DAVID Bioinformatic Resources 6.7 (Huang *et al*, 2009). To get a list of uniquely up-regulated genes in the gene ontology analysis (Figs 1I and 4D), BH-corrected *P*-values < 0.001 were used to get the genes strongly up-regulated in one group (fetal fibroblasts + pB.pA in Fig 1I and pB.pA + RESTi in Fig 4D), while genes with *P*-value < 0.05 in the other group (adult fibroblasts + pB.pA in Fig 1I and pB.mir9/124.pA in Fig 4D) were removed from the gene list. This ensured that no genes that showed a strong trend for

up-regulation were classified as "not up-regulated". For the principal component analysis (PCA), one of the pB.pA + RESTi triplicate clustered with the pB.pA group which is most likely due to lack of co-expression of pB.pA and *REST* shRNA as they are delivered on separate vectors. This group was excluded from further analysis.

### Data availability

The RNAseq dataset can be found on the GEO repository under accession number GSE90068.

### Transplantation

Adult fibroblasts were first transduced with Syn-GFP and then with lentiviral vectors containing pB.pA, REST shRNAs. Cells were prepared for transplantation 3 days post initiation of neural conversion and transplanted to the striatum of male and female neonatal Sprague Dawley rats (p1; Charles River) under hypothermia anesthesia using a 5-µl Hamilton syringe fitted with a glass capillary (outer diameter 60–80 µm). The rats received a 1-µl injection of 100,000 cells through one needle penetration[‡]. After injection, the syringe was left in place for 2 min before being retracted slowly. Animals were housed in standard cages, under a 12-h light/dark cycle with *ad libitum* access to food and water. All procedures were conducted in accordance with the European Union Directive (2010/63/EU), were approved by the ethical committee for the use of laboratory animals at Lund University and the Swedish Department of Agriculture (Jordbruksverket), and were performed in compliance with the ARRIVE guidelines.

### Electrophysiology

*In vitro* patch-clamp electrophysiology was performed on iNs reprogrammed from adult dermal fibroblasts on coverslips and co-cultured with glia between days 85 and 100 post-transduction. Cells were recorded in a Krebs solution composed of (in mM): 119 NaCl, 2.5 KCl, 1.3 $MgSO_4$, 2.5 $CaCl_2$, 25 glucose, and 26 $NaHCO_3$. Cells ($n = 20$) with a neuronal morphology as evidenced by them possessing a round cell body, processes and expressing GFP under the control of the synapsin promoter (co-transduced with the reprogramming factors) were patched for whole-cell recordings.

For recordings on slices, coronal brain slices from transplanted rats were prepared at 8 weeks postconversion. Rats were killed by an overdose of pentobarbital, and the brains were rapidly removed and cut coronally on a vibratome at 275 µm. Slices were transferred to a recording chamber and submerged in a continuously flowing Krebs solution gassed with 95% $O_2$ and 5% $CO_2$ at 28°C. The composition of the Krebs solution for slice recording was (in mM): 126 NaCl, 2.5 KCl, 1.2 $NaH_2PO_4$-$H_2O$, 1.3 $MgCl_2$-$6H_2O$, and 2.4 $CaCl_2$-$6H_2O$. Converted cells were identified by their GFP fluorescence and patched ($n = 8$ in total).

Recordings were made using Multi-clamp 700B (Molecular Devices), and signals were acquired at 10 kHz using pClamp10 software and a data acquisition unit (Digidata 1440A, Molecular Devices). Borosilicate glass pipettes (3–7 MΩ) for patching were

### The paper explained

**Problem**
Direct neural reprogramming holds great promises for disease modeling and cell-based replacement therapy for neurodegenerative disorders. However, no current reprogramming approach is sufficiently efficient to allow the use of this technology using patient-derived material for high content biomedical applications.

**Results**
We provide mechanistic insights and a new strategy for direct neuronal reprogramming specifically adapted for the conversion of dermal fibroblasts of elderly donors, including those derived from patients with neurodegenerative disorders such as Alzheimer's, Parkinson's, and Huntington's diseases.

**Impact**
Our new one vector conversion system offers new and important opportunities to obtain patient- and disease-specific neurons for disease modeling, drug screening, diagnostics, and transplantation.

filled with the following intracellular solution (in mM): 122.5 potassium gluconate, 12.5 KCl, 0.2 EGTA, 10 Hepes, 2 MgATP, 0.3 $Na_3GTP$, and 8 NaCl and adjusted to pH 7.3 with KOH as in (Pfisterer *et al*, 2011a). Resting membrane potentials were monitored immediately after breaking into the cell, in current-clamp mode. In cultures, cells were kept at a membrane potential of −60 to −80 mV, and 500 ms currents were injected from −20 pA to +90 pA using 10 pA increments to induce action potentials. For slices, action potentials were induced with a 500 ms current injected from −100 pA to +400 pA with 50 pA increments. Spontaneous postsynaptic activity was recorded in current-clamp mode at resting membrane potentials using 0.1 kHz lowpass filter.

### Statistical analysis

All data are expressed as mean ± the standard error of the mean. Whenever the analysis is performed with one cell line, biological replicates ($n = 3–4$) were used. In case of experiments using multiple cell lines, we used $n = 5$ to account for inter-individual variation. For electrophysiology on slices, we estimated that $n = 2$ neurons would be recorded per animal and animals could not be randomized nor done blind as they all received the same cell suspension. A Shapiro–Wilk normality test was used to assess the normality of the distribution. When a normal distribution could not be assumed, a nonparametric test was performed. Groups were compared using a one-way ANOVA with a Bonferroni *post hoc* or a Kruskal–Wallis test with a Dunn's or Conover multiple comparisons tests. In case of only two groups, they were compared using a Student's *t*-test. An *F*-test was used to compare variance, and in case of unequal variance, a Welch's correction test was then performed. Statistical analyses were conducted using the GraphPad Prism 7.0. An alpha level of $P < 0.05$ was set for significance.

**Expanded View** for this article is available online.

---

[‡]Correction added on 1 August 2017 after first online publication: The mode of anesthesia and number of cells injected was corrected.

## Acknowledgements

We thank Marie Persson Vejgården and Sol Da Rocha Baez for technical assistance as well as Dr. Anna Hammarberg for her valuable help with high content screening and FACS experiments and Dr. Andreas Heuer for help with neonatal injections. The research leading to these results has received funding from the New York Stem Cell Foundation, the European Research Council under the European Union's Seventh Framework Programme: FP/2007-2013 Neuro Stem Cell Repair (no. 602278) and ERC Grant Agreement no. 30971, the Swedish Research Council (grant agreement 521-2012-5624, 2016-00873 and 70862601/Bagadilico), Swedish Parkinson Foundation (Parkinsonfonden), and the Strategic Research Area at Lund University Multipark (multidisciplinary research in Parkinson's disease). Janelle Drouin-Ouellet is supported by a Canadian Institutes of Health Research (CIHR) fellowship (#358492), Johan Jakobsson is supported by the Swedish Foundation for Strategic Research (# FFL12-0074), and Roger Barker is supported by an NIHR Biomedical Research Centre grant to the University of Cambridge/Addenbrooke's Hospital. We would also like to acknowledge the regional agreement on medical training and clinical research (ALF) between Stockholm County Council and Karolinska Institutet. Malin Parmar is a New York Stem Cell Foundation—Robertson Investigator.

## Author contributions

JD-O, SL, JJ, and MP designed research; JD-O, SL, DRO, KP, DAG, LMC, and RV performed research; AAS, GW-T, CG, LM, HT, and RAB contributed new reagents/analytic tools; JD-O, SL, P-LB, and DRO analyzed data; JD-O, JJ, and MP wrote the paper.

## Conflict of interest

The authors declare that they have no conflict of interest.

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
