## [Review Process File · EMBO Molecular Medicine]

REST suppression mediates neural conversion of adult human fibroblasts via microRNA-dependent and -independent pathways

Janelle Drouin-Ouellet, Shong Lau, Per Brattås, Daniella Ottosson, Karolina Pircs, Daniela Grassi, Lucy Collins, Romina Vuono, Annika Andersson Sjöland, Gunilla Westergren-Thorsson, Caroline Graff, Lennart Minthon, Håkan Toresson, Roger Barker, Johan Jakobsson, and Malin Parmar

Corresponding author: Malin Parmar, Lund University

Review timeline:

Submission date:	16 December 2016
Editorial Decision:	17 January 2017
Author's Appeal:	06 March 2017
Editorial Decision:	24 April 2017
Revision received:	12 May 2017
Accepted:	18 May 2017

Transaction Report:

Editor: Céline Carret

1st Editorial Decision

17 January 2017

Thank you for the submission of your manuscript "Highly efficient one-step generation of induced neurons from patients with chronic neurodegenerative diseases using a novel vector system" and please accept my apologies for not replying earlier due to a combination of events: Christmas holidays, difficulties in securing three and willing referees leading to post-review asking for editorial advice, and the time to get the latter. We have now heard back from the two referees and our advisor.

You will see that while referee 1 is rather supportive, referee 2 (who knows our journal well) is not and based this evaluation on the nature of the study i.e. mainly methodological. Unfortunately this assessment was shared by our advisor who stated "This is clearly a technical manuscript reporting an improvement of an existing method. [...] Although the manuscript reports a methodological improvement, the advance is purely technical and somewhat incremental. In addition, there is no clear evidence of scientific insights that can be obtained using the new method but could not be achieved before. Thus, I would have doubts as to whether EMM is the right journal for this."

I hope you can see that while the study is considered to be robust, these evaluations preclude further consideration for publication in EMBO Molecular Medicine. I hope that the reports will allow you to select a better-suited venue without delay.

I am sorry to have to disappoint you on this occasion, and hope that this negative decision does not

prevent you from considering EMBO Medicine for the publication of future studies.

***** Reviewer's comments *****

Referee #1 (Remarks):

Drouin-Ouelet et al report the generation of a single vector system that expresses pro-neurogenic factors (Brn2 and Ascl1) and shRNA against REST (RESTi) in order to improve the neuronal conversion of elderly fibroblasts. First, authors systemically probed the neuronal conversion efficiency by expressing Brn2 and Ascl1 in lentiviral vectors with a different sequential expression of the genes, and with or without WPRE element. Authors found that a vector so-called pB.pA expressing Brn2 under PGK promoter and Ascl1 under a separate PGK promoter is highly efficient in neuronal conversion. Furthermore, authors demonstrated that inhibition of REST expression via shRNA further increased the neural conversion efficiency in combination with pB.pA, which generate neurons from elderly fibroblasts in high efficiency and purity. Mechanistically, authors found that the induction of miR-9 is partly responsible for the improved neuronal conversion when RESTi is added. Overall, the manuscript is well written, and shows the thoughtful design of experiments, reporting one of the most efficient ways of performing a direct neuronal conversion of fibroblasts. Below are comments to improve the manuscript.

1. In measuring efficiency and purity of neural conversion in Figure 2C and 2D, how did authors measure these? Description in text is confusing. How are % of MAP+ or TAU+ cells different in measuring the efficiency vs. the purity?
2. In Figure 2F and 2G that show the presence of Syn-GFP positive neurons, it is not described how authors labeled cells. Information to the vector, at which stage cells were labeled, etc is missing.
3. In Figure 3E, authors need to show how the statistic analysis was performed. How many experiments were performed, which cells, etc?
4. In Figure 5E that shows the improved neuronal conversion of HD line, authors need to give a specific description of how they optimized the condition rather than writing that you just optimized it. Then, authors need to add it to Figure 5D to make comparison with efficiency in other lines as well as HD neuronal conversion of their standard method.

Referee #2 (Remarks):

This manuscript by Drouin-Ouellet et al., describes a method for high-yield direct conversion of human fibroblasts into neurons using a dual promoter-based vector that co-delivers reprogramming factors alone or in combination with neuron specific micro RNAs or REST-silencing transcription factors. Although several variants of these methods have been published, what is new in this paper is the use of reprogramming factors in combination with REST-silencing in the same construct. This is a pure methodological paper and, therefore, I have doubts that EMBO Mol Med is the right place to publish it. In the last section of the manuscript the authors report the generation of neurons from fibroblasts of Parkinson (sporadic and genetic), Alzheimer and Huntington patients, however no functional characterization was attempted. Neurons generated from patients were not investigated to see if they show manifestations of the diseases.

Additional comments

1) Page 12. The authors wrote "confirming that a bicistronic approach is preferable as it delivers both conversion factors to each cell and reduces the variability of transgene expression in the overall cell population"

This assumption does not appear to correlate with the images shown in figure 1d: there are few Brn2a + cells (high expression) and many more Ascl1+ cells when using pB.pA lentivirus: it seems there is high variability of transgene expression among the cells.

2) The authors establish comparisons between fibroblasts from different tissues (lung and dermis). It should have been preferable to use the same type of fibroblast to perform the experiments:

Figures 1h, 1i, 1j: foetal lung fibroblasts vs adult dermal fibroblasts.

Figure 2a: qPCR analysis of REST gene expression in foetal lung fibroblasts vs adult dermal fibroblasts.

In figure 2b the authors used adult lung fibroblast to induce neuron conversion. As the authors already have this sample, it should have been used for the analyses in figures 1h, 1i, 1j and 2a to compare adult and foetal lung fibroblasts.

3) It is not clear to me that the authors convincingly demonstrate that effect of REST-silencing inducing a high yield of neuron conversion is partially mediated via up-regulation of miR-9/9* and miR-124.

Reprogramming with pB.pA lentivirus increases the miRNA levels at a higher magnitude than the differences observed between the conditions with or without REST inhibition. It is true that the expression level of a microRNA can be determinant for its function. Then, authors should analyze the level of expression of the mature miRNAs after transduction with the miR9/9*-miR-124 construct to confirm that the levels of expression are similar to the ones observed with RESTi reprogramming and higher than after reprogramming with pB.pA only.

In addition, it would be interesting to analyze the miRNA levels of expression in fetal versus adult fibroblasts after reprogramming without REST inhibition, to see whether the differences in the yield of neuron conversion correlate with the level of miRNAs expression.

4) Figures 3d and 3e. Why is the % of TAU+ cells so low in the control (CTR) conditions? This value, that in theory represents the neuronal conversion yield, is considerably lower than the values the authors show in other figures (for example, fig 3c: neuronal yield based on MAP2 expression).

5) The paper is in general technically fine, although the quality is low in some cases. For example there seems to be something wrong with the electrical recording in Fig 2f where in the different sweeps the baseline changes more than 20 mV.

Authors' Appeal

06 March 2017

Thank you for agreeing to reconsider our manuscript formerly entitled "Highly efficient one-step generation of induced neurons from patients with chronic neurodegenerative diseases using a novel vector system", which is now entitled "REST inhibition overcomes difference in reprogramming requirements between fetal and adult human fibroblasts".

Firstly, we would like to address the issue about novelty that was raised as a concern by one of the reviewers. While other studies have investigated the differences in reprogramming requirements between primary and cultured astrocytes (Masserdotti et al., 2015), and between mouse and human fibroblasts (Xue et al., 2013; Xue et al., 2016), this is the first study that specifically investigates the difference in reprogramming requirements between fetal and adult human fibroblasts. Independent of the previous studies, we have identified the REST complex as an important barrier to reprogramming of adult human fibroblasts.

This is the first study that presents global gene expression profiling comparing the transcriptional response in fetal vs. adult human fibroblasts, and shows that the most commonly used neural conversion genes (Ascl1 and Brn2) elicit largely distinct transcriptional responses between these two populations. This dataset points to REST as a potential reprogramming barrier in adult fibroblasts and we show that indeed REST inhibition (RESTi) can remove this reprogramming barrier. Via functional experiments, whereby we overexpress and knockdown the neuron specific microRNAs miR-9 and miR-124, we show that the effect of RESTi during conversion of adult fibroblasts is mediated in part via microRNA up-regulation, but also through microRNA independent mechanisms. Furthermore, we investigate the difference in neural cell fate acquisition at the transcriptional level between RESTi, microRNA and RESTi + microRNA, which confirms that RESTi activates a similar subset of neuronal genes as microRNA overexpression and additionally, a distinct set of neuronal genes not activated via microRNA overexpression alone. Based on these findings, we design an all-in-one conversion vector and show its utility in a number of fibroblasts collected from three different clinical sites, representing 5 types of neurological disease and donor ages of 52 to 77 years old. Our findings now make it possible to generate induced neurons of sufficiently high numbers to allow large scale biomedical applications such as disease

modeling, drug screening and early and differential diagnostics. In addition, the cells may represent a possibility for personalized cell based therapies. This is a significant advancement in the field, well within the scope of *EMBO Molecular Medicine* that publishes papers at the interface between clinical research and basic science, and should have a broad appeal for the readers of the journal.

We have now made substantial changes to the manuscript in order to better highlight these novel mechanistic insights. In addition, we have also addressed the reviewer comments as detailed in the point-by-point below.

***** Reviewer's comments *****

Referee #1 (Remarks):

Drouin-Ouellet et al report the generation of a single vector system that expresses pro-neurogenic factors (Brn2 and Ascl1) and shRNA against REST (RESTi) in order to improve the neuronal conversion of elderly fibroblasts. First, authors systemically probed the neuronal conversion efficiency by expressing Brn2 and Ascl1 in lentiviral vectors with a different sequential expression of the genes, and with or without WPRE element. Authors found that a vector so-called pB.pA expressing Brn2 under PGK promoter and Ascl1 under a separate PGK promoter is highly efficient in neuronal conversion. Furthermore, authors demonstrated that inhibition of REST expression via shRNA further increased the neural conversion efficiency in combination with pB.pA, which generate neurons from elderly fibroblasts in high efficiency and purity. Mechanistically, authors found that the induction of miR-9 is partly responsible for the improved neuronal conversion when RESTi is added. Overall, the manuscript is well written, and shows the thoughtful design of experiments, reporting one of the most efficient ways of performing a direct neuronal conversion of fibroblasts. Below are comments to improve the manuscript.

We thank the reviewer for positive assessment of our study and for the comments suggested to improve the manuscript.

1. In measuring efficiency and purity of neural conversion in Figure 2C and 2D, how did authors measure these? Description in text is confusing. How are % of MAP+ or TAU+ cells different in measuring the efficiency vs. the purity?

We chose to quantify the conversion efficiency and purity of neural conversion according to Vierbuchen et al., (2010) and Ladewig et al. (2013). This means that the neuronal purity was calculated as the number of MAP2⁺ cells over the total number of cells in the well at the end of the experiment. In some cases, we use also Tau as a neuronal marker in addition to MAP2 in order to assess the conversion into more mature neurons. Conversion efficiency was calculated as the number of TAU⁺ over the total number of fibroblasts plated for reprogramming. Together, these two metrics gives a good insight into how efficient the neural conversion is. We have now reported this more clearly in the manuscript on page 9 and a description of the calculations has been added to the method section "*Immunocytochemistry, imaging and high content screening quantifications*" on page 14.

2. In Figure 2F and 2G that show the presence of Syn-GFP positive neurons, it is not described how authors labeled cells. Information to the vector, at which stage cells were labeled, etc is missing.

For the electrophysiological recordings, cells were transduced with a vector generated in-house that expresses GFP under the control of the neuron specific *Synapsin* promoter in order to identify the human iNs in the rat brain after transplantation. We have now better described this in the results section on page 6 and details about the *Syn-GFP* vector have been added to the method sections "*Viral Vectors and Virus Transduction*" on page 13 and "*Electrophysiology*" on page 16.

3. In Figure 3E, authors need to show how the statistic analysis was performed. How many experiments were performed, which cells, etc?

We performed the experiment in triplicate and analyzed the data with a Student *t* test. This has been added to the method section "*qRT-PCR analysis for miR-9, miR-124 and RE1-*

silencing transcription factor" on page 15.

4. In Figure 5E that shows the improved neuronal conversion of HD line, authors need to give a specific description of how they optimized the condition rather than writing that you just optimized it. Then, authors need to add it to Figure 5D to make comparison with efficiency in other lines as well as HD neuronal conversion of their standard method.

To improve the conversion of the HD line, we omitted the re-plating step on day 12 which increased the cell survival and prevented the cells from detaching. This resulted in a better yield and purity of the iNs from this line. We have now quantified the HD lines converted with the new protocol (MAP2+ neuronal count 13857 ± 250 ; purity $62.4 \pm 3.4\%$) which is actually higher than the other lines. This has been included on page 9.

Referee #2 (Remarks):

This manuscript by Drouin-Ouellet et al., describes a method for high-yield direct conversion of human fibroblasts into neurons using a dual promoter-based vector that co-delivers reprogramming factors alone or in combination with neuron specific micro RNAs or REST-silencing transcription factors. Although several variants of these methods have been published, what is new in this paper is the use of reprogramming factors in combination with REST-silencing in the same construct. This is a pure methodological paper and, therefore, I have doubts that EMBO Mol Med is the right place to publish it. In the last section of the manuscript the authors report the generation of neurons from fibroblasts of Parkinson (sporadic and genetic), Alzheimer and Huntington patients, however no functional characterization was attempted. Neurons generated from patients were not investigated to see if they show manifestations of the diseases.

We thank the reviewer for helpful and constructive comments on our manuscript that we have responded to below. We do also acknowledge that the way the data was presented in the first version of the manuscript could give the impression of a methodological paper. However, as we describe in the letter above, this is the first study that:

- Presents global gene expression profiling comparing the transcriptional response in fetal vs. adult fibroblasts and shows that the most commonly used neural conversion genes (Ascl1 and Brn2) elicit largely distinct transcriptional responses between these two populations.
- Via functional experiments, where we overexpress and knockdown the neuron specific microRNAs miR-9 and miR-124, shows that the effect of RESTi during conversion of adult fibroblasts is only partially mediated via microRNA up-regulation.
- Investigates the difference in neural cell fate acquisition at the transcriptional level between RESTi, microRNA and RESTi + microRNA, which confirms that RESTi activates a similar subset of neuronal genes as microRNA overexpression and additionally, a distinct set of neuronal genes not activated via microRNA overexpression alone.

Based on these findings we design an all-in-one conversion vector and show its utility in a number of fibroblast samples collected from three different clinical sites, representing 5 types of neurological disease and donor ages of 52 to 77 years old. Analysis of disease related pathology for each disease is of course both interesting and relevant, but is an extensive study for each disease and falls outside the scope of this study that focuses on a better mechanistic insight into reprogramming requirements of adult vs. fetal fibroblasts in order to improve the reprogramming of adult fibroblasts and rendering it suitable for biomedical applications.

Additional comments

1) Page 12. The authors wrote "confirming that a bicistronic approach is preferable as it delivers both conversion factors to each cell and reduces the variability of transgene expression in the overall cell population"

This assumption does not appear to correlate with the images shown in figure 1d: there are few Brn2a + cells (high expression) and many more Ascl1+ cells when using pB.pA lentivirus: it seems there is high variability of transgene expression among the cells.

The dual promoter vector delivers both transgenes to the cells. However, the level of expression of

each transgene varies between each cell. In general, when assessed with immunocytochemistry using antibodies for ASCL1 and BRN2a, we find that the second transgene is expressed at higher levels as seen in Figure 1d, which has now been modified to better show this.

However, since immunochemical staining depends on the quality of the antibody and is not quantitative, we also made a separate vector with GFP that confirmed that the fluorescent intensity of the GFP is higher in most cells when expressed after the second promoter (Suppl. Fig 1). We have now clarified this in the manuscript on page 4.

2) The authors establish comparisons between fibroblasts from different tissues (lung and dermis). It should have been preferable to use the same type of fibroblast to perform the experiments:

Figures 1h, 1i, 1j: foetal lung fibroblasts vs adult dermal fibroblasts.

Figure 2a: qPCR analysis of REST gene expression in foetal lung fibroblasts vs adult dermal fibroblasts.

In figure 2b the authors used adult lung fibroblast to induce neuron conversion. As the authors already have this sample, it should have been used for the analyses in figures 1h, 1i, 1j and 2a to compare adult and foetal lung fibroblasts.

We agree that comparing fetal and adult fibroblasts from the same tissue is compelling. However, we chose to use fetal lung fibroblasts given that it is a commercially available, standardized and a commonly used fetal fibroblast line. We have used this line to develop and study direct neural conversion previously and therefore also used it when designing and testing the dual promoter vectors.

During this work, it became apparent that specific reprogramming barriers exist in adult human fibroblasts compared to fetal fibroblasts. For these studies, we focused on adult dermal fibroblasts given its accessibility, in order to get direct insights into the molecular biology behind direct reprogramming of adult fibroblasts. However, we did confirm that the new strategy also worked on adult lung fibroblasts to establish that the observed differences was in fact due to age of the donor, and not due to the source of fibroblasts (Fig 2b). We have now also confirmed that RESTi is necessary for conversion of adult lung fibroblasts and this data has been included in Fig 1h.

3) It is not clear to me that the authors convincingly demonstrate that effect of REST-silencing inducing a high yield of neuron conversion is partially mediated via up-regulation of miR-9/9* and miR-124.

Reprogramming with pB.pA lentivirus increases the miRNA levels at a higher magnitude than the differences observed between the conditions with or without REST inhibition. It is true that the expression level of a microRNA can be determinant for its function. Then, authors should analyze the level of expression of the mature miRNAs after transduction with the miR9/9*-miR-124 construct to confirm that the levels of expression are similar to the ones observed with RESTi reprogramming and higher than after reprogramming with pB.pA only. In addition, it would be interesting to analyze the miRNA levels of expression in fetal versus adult fibroblasts after reprogramming without REST inhibition, to see whether the differences in the yield of neuron conversion correlate with the level of miRNAs expression.

It is well known that the microRNA loop we have used results in high-level microRNA expression (Yoo et al., 2011), and we have now also confirmed this using our vector (Suppl. Fig 2). The level of expression and activity of the microRNA is likely to be far higher than the endogenous microRNA expression induced by RESTi in most cells. However, it is not trivial to correlate bulk microRNA expression to response in a cell. Still, a number of experimental observations in our dataset supports that the RESTi effect on neural conversion is partially mediated via up-regulation of microRNAs. We base this on the following experimental observations:

- 1) In both the pB.pA + RESTi and the pB.miR9/124.pA conditions, each cell will express a different level and ratio of conversion genes and microRNAs, yet they reprogram adult fibroblasts at similar efficiencies (Fig 3d)
- 2) We see no added effect on neural conversion efficiency using RESTi when also adding microRNAs (Fig 3d)
- 3) Blocking microRNA activation during neural conversion using RESTi diminishes the conversion (Fig 3 e-g)
- 4) The transcriptional response in pB.pA + RESTi is very similar to pB.miR9/124.pA, and there is no significant effect of combining RESTi with microRNA expression (Figs 4 and

5)

4) Figures 3d and 3e. Why is the % of TAU+ cells so low in the control (CTR) conditions? This value, that in theory represents the neuronal conversion yield, is considerably lower than the values the authors show in other figures (for example, fig 3c: neuronal yield based on MAP2 expression).

In the analysis of the first submission, we used a different and more stringent cut off than what we used in the other figures. We have now redone the analysis using the same settings as we have used in the other figures and the purity is now comparable to those experiments. With both these settings, the outcome of the experiment is the same i.e miR-9 knockdown impairs RESTi mediated conversion. The new data has been inserted to Fig 3e and f.

5) The paper is in general technically fine, although the quality is low in some cases. For example there seems to be something wrong with the electrical recording in Fig 2f where in the different sweeps the baseline changes more than 20 mV

Long-term culturing of iNs for physiological maturation is technically challenging and whole-cell patch in itself disturbs the cells' equilibrium when they have not yet reached a full and stable maturity. This is often seen with an unstable baseline for reprogrammed cells (Lu et al., 2013; Hu et al., 2015). Nevertheless, the ability to fire an action potential can still be determined. We have now presented only the firing trace, which is the relevant one and removed the "baseline traces" as is often presented in publications for more clarity (Pfisterer et al., 2011; Shi et al., 2016).

References

- Hu, W, B Qiu, W Guan, Q Wang, M Wang, W Li, L Gao, L Shen, Y Huang, G Xie, H Zhao, Y Jin, B Tang, Y Yu, J Zhao, and G Pei. 2015. Direct Conversion of Normal and Alzheimer's Disease Human Fibroblasts into Neuronal Cells by Small Molecules. *Cell Stem Cell* 17, no. 2: 204-212.
- Ladewig, J, P Koch, and O Brüstle. 2013. Leveling Waddington: the emergence of direct programming and the loss of cell fate hierarchies. *Nat Rev Mol Cell Biol* 14, no. 4: 225-236.
- Lu, J, H Liu, CT Huang, H Chen, Z Du, Y Liu, MA Sherafat, and SC Zhang. 2013. Generation of integration-free and region-specific neural progenitors from primate fibroblasts. *Cell Rep* 3, no. 5: 1580-1591.
- Masserdotti, G, S Gillotin, B Sutor, D Drechsel, M Irmeler, HF Jørgensen, S Sass, FJ Theis, J Beckers, B Berninger, F Guillemot, and M Götz. 2015. Transcriptional Mechanisms of Proneural Factors and REST in Regulating Neuronal Reprogramming of Astrocytes. *Cell Stem Cell* 17, no. 1: 74-88.
- Pfisterer, U, A Kirkeby, O Torper, J Wood, J Nelander, A Dufour, A Bjorklund, O Lindvall, J Jakobsson, and M Parmar. 2011. Direct conversion of human fibroblasts to dopaminergic neurons. *Proc Natl Acad Sci U S A* 108, no. 25: 10343-10348.
- Shi, Z, J Zhang, S Chen, Y Li, X Lei, H Qiao, Q Zhu, B Hu, Q Zhou, and J Jiao. 2016. Conversion of Fibroblasts to Parvalbumin Neurons by One Transcription Factor, Ascl1, and the Chemical Compound Forskolin. *J Biol Chem* 291, no. 26: 13560-13570.
- Vierbuchen, T, A Ostermeier, ZP Pang, Y Kokubu, TC Sudhof, and M Wernig. 2010. Direct conversion of fibroblasts to functional neurons by defined factors. *Nature* 463, no. 7284: 1035-1041.
- Xue, Y, K Ouyang, J Huang, Y Zhou, H Ouyang, H Li, G Wang, Q Wu, C Wei, Y Bi, L Jiang, Z Cai, H Sun, K Zhang, Y Zhang, J Chen, and XD Fu. 2013. Direct conversion of fibroblasts to neurons by reprogramming PTB-regulated microRNA circuits. *Cell* 152, no. 1-2: 82-96.
- Xue, Y, H Qian, J Hu, B Zhou, Y Zhou, X Hu, A Karakhanyan, Z Pang, and XD Fu. 2016. Sequential regulatory loops as key gatekeepers for neuronal reprogramming in human cells. *Nat Neurosci* 19, no. 6: 807-815.
- Yoo, AS, AX Sun, L Li, A Shcheglovitov, T Portmann, Y Li, C Lee-Messer, RE Dolmetsch, RW Tsien, and GR Crabtree. 2011. MicroRNA-mediated conversion of human fibroblasts to neurons. *Nature* 476, no. 7359: 228-231.

2nd Editorial Decision

24 April 2017

Thank you so much for your incredible patience while we were re-evaluating your revised article for EMBO Molecular Medicine. We have now decided to move forward with publication based on the editorial advices we received. I am pleased to inform you that we will be able to accept your manuscript pending final amendments.

I look forward to reading a new revised version of your manuscript as soon as possible.

1st Revision - authors' response

12 May 2017

Authors made the requested editorial changes.

Corresponding Author Name: Malin Parmar

Manuscript Number: EMM-2016-07471-V3